# A qualitative evaluation of a compassion-focused therapy group intervention for UK healthcare staff at an acute hospital trust

Aikaterini Rammou[1,2]*, Sophie M. Allan[1,3], Martha Dean-Tozer[4], James Baker[4]

1 Department of Clinical Psychology and Psychological Therapies, Norwich Medical School, University of East Anglia, Norwich, United Kingdom, 2 Cambridgeshire and Peterborough NHS Foundation Trust, Cambridge, United Kingdom, 3 School of Health Sciences, University of East Anglia, Norwich, United Kingdom, 4 West Suffolk NHS Foundation Trust, Suffolk, United Kingdom

* rammou.katerina0@gmail.com

## Abstract

Healthcare staff can encounter empathy distress, burnout, and trauma, often responding with self-criticism rather than self-compassion, yet there remains limited evaluation of compassion-focused interventions for staff with mental health difficulties. This study qualitatively evaluated a compassion-focused therapy (CFT) group intervention offered to National Health System (NHS) staff with mental health difficulties and high self-criticism. A qualitative design using semi-structured interviews was used. Eight staff who attended a 12-week CFT group were interviewed about their experiences learning key CFT concepts, group-based learning, the intervention's impact on their well-being, and suggestions for improvement. Reflexive thematic analysis was conducted. Participants found the group beneficial, emphasising the development of group safeness, which fostered connection and vulnerability through shared NHS and caregiver identities. CFT content was seen as highly relevant, addressing self-criticism, while helping participants label personal experiences and enhancing self-awareness. The group format fostered a shared understanding of human suffering, reduced feelings of isolation and widened participants' perspectives through interpersonal learning. Participants reported acquiring skills to support their well-being, with some noting overall emotional improvement. Challenges included emotionally triggering content, highlighting the need for tailored support. Suggestions included smaller group sizes, flexible content delivery to meet individual and group needs, and additional support such as one-on-one sessions or booster sessions. Protected time from work, more information about the group process, and clear expectations were recommended to enhance group safeness and reduce initial hesitation. Group-based CFT was described as acceptable and beneficial, offering preliminary evidence for its potential to support NHS staff with mental health difficulties and pervasive self-criticism. Larger, mixed-method evaluations are recommended to explore the intervention's broader impact and mechanisms.

**Data availability statement:** Data cannot be shared publicly because it contains information collected under NHS service evaluation governance and is subject to the UK Data Protection Act (2018) and the Trust's confidentiality policy. External researchers may submit a formal request to the NHS Trust's Clinical Audit department (ClinicalAudit@wsh.nhs.uk) which would be reviewed by the Trust's Information Governance team and, if appropriate, approved by the Caldicott Guardian before any anonymised data could be shared.

**Funding:** The author(s) received no specific funding for this work.

**Competing interests:** The authors have declared that no competing interests exist.

## Introduction

The health and well-being of National Health Service (NHS) staff have been a long-standing concern, with sickness absence rates higher than in other public and private sectors [1]. Rimmer [2] indicated that about 40% of absences of NHS staff are due to work-related stress whereas a report by NHS England highlighted that around one-third of absences are linked to mental health difficulties [3]. The COVID-19 pandemic intensified work pressures for healthcare professionals [4,5], increasing the risk of adverse mental health outcomes [6]. In response, government initiatives were launched to improve staff well-being, including tailored in-house mental health support [7,8].

Research highlights compassion as a potential antidote to the mental health challenges and work-related stress healthcare professionals face [9]. Compassion, defined as "sensitivity to suffering in self and others with a commitment to alleviate and prevent it" [10 (p.19)], can flow in three directions: self-compassion, receiving compassion from others, and offering compassion to others [10,11]. Studies have found that higher self-compassion is associated with lower levels of burn-out [12] and better well-being [13–15], while lower self-compassion correlates with psychopathology [16].

Fear of receiving compassion, from oneself or others, has been associated with mental health outcomes such as depression, anxiety, and wellbeing [17]. Conversely, fostering compassion can help regulate negative emotions, facilitate responses to distress, and promote feelings of safety [18]. However, past negative experiences may contribute to the development of fears, blocks, and resistances (FBRs) to compassion [19] which can hinder all three flows of compassion [20]. FBRs, particularly toward oneself, have been consistently linked to mental health outcomes and vulnerability factors such as self-criticism [21] and shame [17].

It is important to note that healthcare workers often express compassion toward others, but this does not necessarily translate into being self-compassionate [22]. This may stem from personal blocks tied to mental health issues, as well as organisational cultures that promote self-sacrifice over self-compassion [23–25]. Staff can also present with particular FBRs to compassion [26], potentially limiting the efficacy of common "well-being" initiatives, and leaving staff "stuck" in shame-driven modes of relating to themselves, feeding patterns of self-neglecting striving behaviour.

Current evidence suggests that compassion-focused interventions could help staff foster self-compassion, improving well-being [15]. One such intervention is compassion-focused therapy (CFT; [11]) which can support individuals to reduce self-criticism, develop self-awareness, and adopt a caring approach toward themselves and others [10,11]. Current evidence supports the effectiveness of CFT for several mental health difficulties. A systematic review of 14 studies across clinical and non-clinical samples reported favourable outcomes in mood, compassion, and well-being [27]. A more recent meta-analysis found CFT improved self-compassion, reduced self-criticism, and alleviated depression within clinical populations [28]. Another meta-analysis examining the effectiveness and acceptability of CFT across a

range of mental health difficulties, highlighted that CFT reduces depression and anxiety while boosting self-compassion, with group CFT having significantly more evidence of effectiveness than individual and self-help interventions [29]. Although not specific to CFT, a meta-analysis of six studies showed that compassion training interventions benefit healthcare staff by improving self-compassion [30].

Considering that NHS staff are working in tremendously difficult times with no clear solutions, compassion seems more needed than ever. Nevertheless, there is still surprisingly little focus within healthcare organisations on how to better promote self-compassion, including the experience of staff in existing compassion-focused intervention initiatives [25,31,32]. Therefore, this study aims to contribute to the scarce literature on compassion-focused interventions for staff who are experiencing mental health difficulties by focusing on the lived experiences of NHS staff participating in a CFT group, offering unique insights into the acceptability, perceived impact, and potential for refining compassion-based interventions tailored to this group.

In response to an increase in referrals and recurring themes of self-criticism and low self-compassion identified in initial psychology assessments, a staff support service at an acute hospital Trust in the East of England launched CFT groups for NHS staff. This service is internally commissioned by the hospital Trust, forms part of the larger psychological services department of the hospital Trust and offers psychological care to NHS staff. This includes individual assessments and evidence-based therapies, group interventions and Trust-wide teaching, training and reflective practice sessions, provided by clinical psychologists, specialist psychological therapists and clinical psychology trainees. The clinical work covers a wide range of presentations including anxiety, low mood, trauma, bereavement and psychological debriefs after clinical incidents. Staff can self-refer or be referred by their line managers.

To our knowledge, no previous research has evaluated a CFT group delivered within an NHS staff support service for healthcare professionals experiencing sustained psychological difficulties and high self-criticism. The present study therefore aimed to explore participants' experiences of this intervention, assess its perceived impact on well-being, and gather suggestions for future development of CFT groups in this context.

The research questions were the following:

1) What was the experience of staff who took part in a CFT group?

2) How did the staff experience learning about different concepts of CFT?

3) How did they experience working on CFT concepts as part of a group?

4) If any, what was the impact of being part of the group on their well-being?

5) What are the participants' views on improving the experience of CFT groups in the future?

## Methods

### Design

A qualitative, cross-sectional design with a self-selecting sample was employed, using semi-structured interviews and questionnaires with NHS staff who participated in a CFT group. The study employed a qualitative approach underpinned by a pragmatic orientation, selecting methods suited to inform service development [33] and assess the experienced acceptability of the CFT group in this context [34], producing findings of practical relevance within the applied service setting [35].

### Ethics statement

All participants provided written informed consent to participation. Ethical approval for the project was gained from the Chair of the Faculty of Medicine and Health Sciences Committee and the University of East Anglia (Reference number: ETH2324-0184), and the relevant NHS Trust Clinical Audit department.

## Confidentiality

Given the small sample size and in-house delivery of the service, there was a heightened risk that participants might be identifiable to colleagues. To minimise this risk, all transcripts and quotes were carefully anonymised to remove details that could reveal individual identities, such as specific roles, departments, or personal circumstances. Descriptive labels in the manuscript were kept broad, and demographic information was reported in aggregated form to further protect confidentiality.

## Participants

Inclusion criteria for being offered a place in the CFT group were the following: (1) being referred to and having an active episode of care with the staff support service; (2) having undergone a formal assessment and formulation with a clinician; and (3) experiencing significant and enduring self-criticism as a main feature of their presenting psychological difficulties. Staff who met these criteria and (4) presented with a sustained psychological difficulty, incorporating work-related stress, anxiety, depression, trauma and Post-Traumatic Stress Disorder (PTSD), or complex mental health difficulties in a stable phase, were invited to take part in the CFT group.

Exclusion criteria for being offered a place at the CFT group were: 1) presenting with significant risk of harm to themselves which would also exclude them from the service; 2), not being able to commit to at least 10 of the 12 sessions; 3) not being able to commit to in-person attendance; and/or 4) undergoing another period of psychological therapy at the same time or having completed a period of psychological therapy within last 3 months.

A total of 20 NHS staff participated in the first CFT three groups, that started in October 2022, January 2023, and May 2023, N = 6, N = 8 and N = 6, respectively. Fifteen (75%) were eligible to take part in the study as they had completed at least eight sessions, N = 6, N = 5 and N = 4, respectively for the three groups. One was on maternity leave so was no longer reachable. All eligible participants were invited via emails from the staff support service, containing a link to the participant information sheet (PIS) and consent form on the Qualtrics secure online survey platform. Participation was voluntary and did not impact the care they received from the service. See Fig 1 below for the recruitment flow diagram.

Eight staff signed up and all took part in the evaluation. A qualitative approach was chosen as this was the first evaluation of the CFT group within the service, primarily aiming to inform service development and assess acceptability. This method was suited to capturing in-depth insights into participants' experiences, particularly given the small sample size. Guidelines recommend six to 10 participants for small interview-based projects [36] thus the sample size of eight was deemed appropriate.

## Compassion-focused therapy group intervention

The compassion-focused group intervention consisted of 12 weekly group sessions. At its outset, the group was based on material from *The Compassionate Mind Workbook* [37] for specific compassionate mind training resources, the *Compassion Focused Therapy: Clinical Practice and Applications* [38] for general CFT practice materials, with a focus on the chapter "Compassion focused therapy in groups" [39]. In the first group, the facilitator based the intervention on the specific chapter from Gilbert and Simos [38], with supporting content from Irons and Beaumont [37] for the Compassionate Mind Training exercises and components (e.g., soothing rhythm breathing, compassionate self, multiple emotional selves practices). At the release of Cattani et al. [40], fidelity to this specific expanded group manual was checked and the group was unchanged, remaining consistent throughout the groups sampled within this evaluation.

The focus of the CFT group intervention was on experiences of persisting self-criticism and shame and working through participants' FBRs to compassion. Sessions incorporated Compassionate Mind Training exercises, with each session beginning with the soothing rhythm breathing exercise and including compassionate imagery exercises. Participants practiced exercises in sessions first and then they were sent weekly emails summarising the

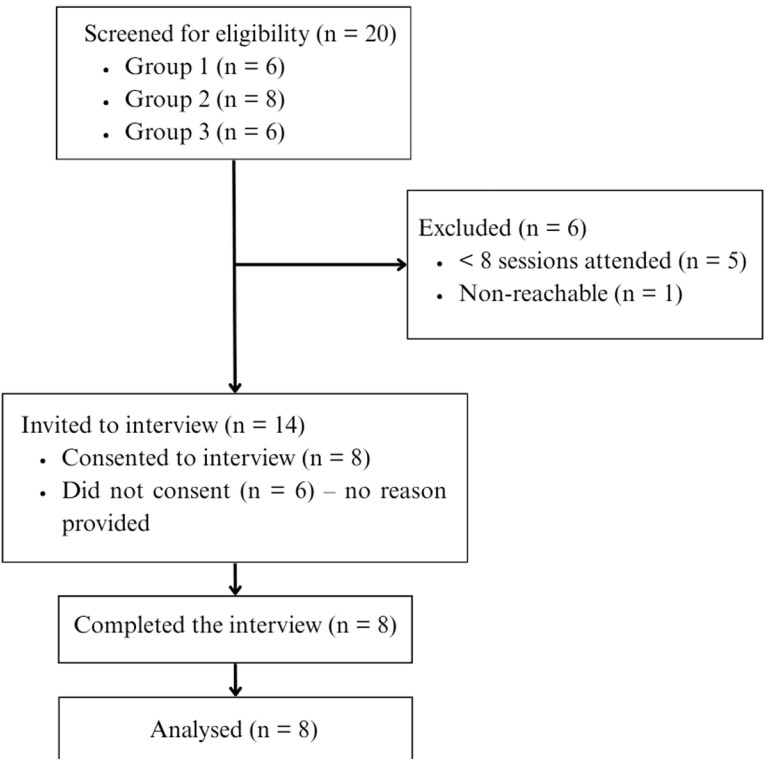

**Fig 1. Recruitment flow diagram.**

session content, with recorded audio practices. CFT content was delivered flexibly and related to common group experiences and was prominently displayed throughout via a projector aiding topic orientation, understanding and retention.

The group facilitator was a clinical psychologist with intermediate experience of CFT practice, in individual and group settings, having attended several foundational and intermediate CFT training seminars on a variety of areas.

## Procedure

NHS staff who provided their written informed consent were directed to an online survey requesting key background information and their preferred contact details. The main researcher invited participants to either an online or in-person interview depending on the participants' preference. Interviews took place from 14th January to 7th of March 2024. Seven interviews were conducted by the evaluating author and one by a trainee clinical psychologist on placement with the service. Support resources were listed on the PIS and in a debriefing email, in case participants required further support. Additional quantitative data was used from pre-intervention questionnaires on mental health symptoms routinely collected by the service and were accessed during the project, from the 14th January to the 7th of March 2024. The first and last authors had access to information that could identify individual participants during or after data collection as they were members of the clinical team the project took place. The remaining authors were involved only after the data had been anonymised.

Interviews were a mean of 63 minutes (range = 47–88 minutes). Seven interviews took place via video-calls on Microsoft Teams, and one was conducted in person.

## Measures

**Demographics and background measures.**  Information on participant's age, gender identity, main NHS job role and the number of CFT group sessions they attended was collected via a brief online questionnaire. Pre-existing service data on the reason for referral and referral source (self-referral or other) were also collected.

**Mental health measures.**  Data on depression and anxiety symptoms as well as self-criticism and self-reassurance were collected prior to starting the CFT group and were used solely to describe participants' baseline clinical characteristics.

The Patient Health Questionnaire-9 (PHQ-9; [41,42]), a 9-item self-report measure, was used to screen for depression and to capture depressive symptom severity with an overall score, with higher scores indicating higher levels of depressive symptoms. The Generalized Anxiety Disorder-7 (GAD-7; [43]), a 7-item self-report measure, was used to capture the presence and severity of symptoms of generalised anxiety in the past two weeks, using the sum of its items, with higher scores representing higher levels of anxiety. The Forms of Self-Criticising/Attacking and Self-Reassuring Scale (FSCRS; [44]), a 22-item self-report measure, was used to capture forms in which people engage in self-attacking behaviours when things are difficult for them. It includes the following subscales: the inadequate self (IS), the hated self (Hs) and the reassured self (RS).

**Semi-structured Interview.**  An interview schedule was collaboratively created by the clinical psychologist who facilitated the CFT groups and the main researcher (for details see S1 File).

The development of the interview schedule was informed to address the following three key domains:

1. CFT theoretical components, including self-compassion, fears/blocks/resistances to compassion, self-criticism and shame, and the "multiple selves" concept, which formed the core focus of the intervention content [11,37,39].

2. Group therapy process factors with a particular emphasis on interpersonal learning, as described in established group psychotherapy literature [45].

3. Intervention evaluation principles focusing on experienced acceptability [34], perceived impact, and recommendations for improvement to inform ongoing service development [46].

The questions were designed to elicit participants' experiences of the CFT group, both in terms of learning and applying CFT concepts and participating in the group format, as well as to capture any perceived impacts on their well-being and suggestions for future delivery of the groups.

## Data analysis

Demographic, background, and clinical data were entered into SPSS [47] to provide descriptive statistics.

Interviews were audio-recorded and transcribed verbatim by the evaluating author into Microsoft Word documents and then entered into the NVivo software [48] for analysis. Reflexive thematic analysis was undertaken following the interview data collection according to Braun and Clarke [49,50], as outlined in Table 1. Reflexive thematic analysis was considered suitable because it focuses on the description of divergence and convergence of experiences rather than on the interpretation of personal experiences or theory. The analytical approach combined inductive and deductive elements [49,50]. Coding initially proceeded inductively, allowing themes to be constructed from participants' accounts without imposing predetermined theoretical categories. While coding began inductively, the organisation of the analysis was also guided by the predefined evaluation questions, which shaped the grouping of codes to ensure alignment with the evaluation aims. Interpretation was subsequently informed by CFT theory [10,11] which was used to contextualise the findings within existing evidence. The analysis followed a semantic approach, focusing on explicit data meaning [50,51].

**Rigour and reflexivity.**  This evaluation adopted a critical realist epistemological stance, acknowledging the influence of researcher perspectives on findings [52]. Given the inherent subjectivity in qualitative research, it is not feasible to

**Table 1. Reflexive thematic analysis steps [49,50].**

| Analysis step | Description |
|---|---|
| Data familiarisation | Interviews and transcription were completed by the main researcher. Individual transcripts were read several times. Initial reflections and observations were noted down, forming initial analytic ideas. |
| Generating initial codes | Initial codes of all transcripts were created highlighting data that were pertinent to the evaluation questions, forming lists of codes between transcripts. Two transcripts were also coded by an assistant psychologist and discrepancies were discussed. |
| Searching for themes | The code list was reviewed. Codes that were too broad or lacked context and codes that were too narrow were renamed. The codes were grouped based on similarity to form potential themes, and structured based on the research questions. |
| Reviewing themes | Themes were re-examined against the data extracts coded to them, assessing their coherence and consistency. Transcripts and themes were reviewed by the group facilitator leading to the reorganisation of some of the themes. |
| Defining and naming themes | Final thematic structure was confirmed. Descriptions for each theme, capturing the essence of what each theme represents were created. |
| Producing the report | Themes were arranged into a narrative organised by the research questions. |

pinpoint the exact impact of the evaluation team's attributes on the conclusions reached [53]. This aligns with a critical realist perspective, which cautions against interpreting research findings as objective reflections of reality [54].

The main researcher had previously been clinically supervised by the group facilitator during a placement at the Staff Support service, which may have introduced a power dynamic and risk of bias in both data collection and interpretation. Specifically, the existing supervisory relationship could have influenced the way interview questions were asked, affected participants' willingness to share critical feedback if they perceived the interviewer as being aligned with the facilitator, and shaped the researcher's interpretation of the data through unconscious positive bias toward the intervention or the facilitator's delivery. This potential influence was explicitly acknowledged and discussed within the research team throughout the study, and the interview schedule deliberately included open, exploratory questions and prompts for both positive and negative feedback. To further mitigate potential bias, the main researcher was not involved in delivering the CFT groups. Participants were also informed that the researcher was independent from the clinical team at the time of the evaluation and that their feedback would not affect their care. For analysis, an independent assistant psychologist from another service double-coded two transcripts. Differences were resolved collaboratively to enhance coding reliability. Additionally, an audit trail in NVivo documented the development of codes and themes, ensuring a transparent and systematic analytical process.

To ensure explicit and comprehensive reporting, the evaluation followed the Consolidated Criteria for Reporting Qualitative Research checklist [55] from the inception of the project (S1 Table).

## Results

### Sample characteristics

Out of the eight participants, three took part in the first CFT group, three in the second and two in the third group. All participants were female, having a White British or White European ethnic background, with mean age of 43.88 ($SD = 9$). All participants were self-referred to the Staff Support service.

At entry to the group, scores on PHQ-9 ranged from 9 to 20 (M = 13, $SD = 4$) representing presentations from mild to severe depression and scores on GAD-7 from 5 to 21 (M = 12.38, $SD = 5.45$) indicating experiences ranging from mild to

severe anxiety. The FSCRS scores on the inadequate self ranged from 22 to 35 (M = 26.63, *SD* = 4.47), on the hated self from 4 to 15 (M = 7.88, *SD* = 3.68) and on the reassured self from 9 to 18 (M = 12.5, *SD* = 3.07).

Table 2 presents participants' job roles, number of sessions attended and reasons for referral to the Staff Support service.

## Thematic analysis findings

An overview of the thematic structure for each evaluation question is outlined in Table 3 below. ID and number of participants endorsing each theme/subtheme with example quotes, can be seen S2 Table.

1. What was the experience of staff who took part in the CFT group?

**Group acceptability.** Overall, the group was perceived as beneficial by all participants. For example, Participant 5 noted it was "*perfect*" for their needs.

*Affiliative experience.* All participants reported a strong sense of connection within the group. They appreciated the caring environment, mutual support, and respect. For example, Participant 7 noted, "*We're all very supportive of each other*". Seven participants spoke about how their shared identity as NHS professionals contributed to a sense of mutual understanding, as they shared similar work-related pressures, experiences, and values. Participant 1 remarked on the shared caregiver identity: "*There was a shared understanding of what it means to be a caregiver*". Participant 6 appreciated the common ground: "*Being NHS staff together actually was quite helpful in the sense that […] we all work hard to look after other people*". Participants 3 and 5 highlighted that, despite holding different professional roles, people in the group had the same workplace with "*the same culture throughout*" (Participant 3), which helped in forming a strong bond. This bond extended beyond the sessions, with six participants mentioning efforts to stay connected after the group ended.

*Developing group safeness.* All participants talked about feeling psychologically safe within the group, allowing them to be vulnerable and authentic. Participant 1 noted, "*you felt you could be you, rather than putting on a front up*", while Participant 7 said, "*I had somewhere to be authentically me and safe*". This safety enabled emotional openness, as seen in Participant 6's experience: "*I felt quite able to show my emotions and say, actually I'm really struggling here*". Despite these positive experiences, seven participants expressed initial tentativeness about joining the group, which gave way to a sense of safeness as the course went on for all of them. Concerns about confidentiality, group dynamics, and managers'

**Table 2. Participant background characteristics (N = 8).**

| Participant number | Job role | No of sessions attended | Reason for referral |
|---|---|---|---|
| Participant 1 | Manager | 12 | Heightened anxiety, with excessive worry and catastrophic ruminative thinking. |
| Participant 2 | Nurse | 8 | Anxiety and depression. Persistently self-critical. |
| Participant 3 | Manager | 10 | Low self-esteem |
| Participant 4 | Allied Health Professional | 12 | Severe self-criticism, shame, and PTSD. |
| Participant 5 | Nurse | 12 | PTSD, significant personal shame and persisting self-criticism. |
| Participant 6 | Allied Health Professional | 11 | Traumatic experiences characterised by feelings of guilt and shame. |
| Participant 7 | Allied Health Professional | 11 | High levels of stress and patterns of overwork leading to burnout, and symptoms of moderate depression and anxiety. |
| Participant 8 | Clinical support staff | 10 | Traumatic childhood experiences. Severe self-criticism, lifelong patterns of prioritising the needs of others. |

**Table 3. Description of themes and subthemes corresponding to each research question.**

| Research question | Themes and subthemes | Description |
|---|---|---|
| What was the experience of staff who took part in the CFT group? | 1. **Group acceptability** | Aspects of the group experience that refer to its acceptability. |
| | 1.1. Developing group safeness | Despite initial tentativeness, the group became a psychologically safe space, fostering vulnerability and authenticity. |
| | 1.2. Affiliative experience | A sense of belonging, support, and respect felt within the group, strengthened by shared NHS background and experiences. |
| | 1.3. Unexpected challenges | Unforeseen difficulties, that participants encountered during sessions, such as personal triggers or emotional discomfort. |
| How did the staff experience learning about different concepts of CFT? | 1. **Usefulness of engaging with key CFT content** | Various aspects valued by participants in relation to learning about CFT content. |
| | 1.1. Helped make sense of personal experiences | CFT content helped participants understand their inner experiences and reduce self-blame by normalising emotional responses, mainly by offering biopsychological and evolutionary frameworks. |
| | 1.2. Building a helpful definition self-compassion | Participants developed a nuanced and more accurate understanding of self-compassion. |
| | 1.3. Increasing self-awareness | Participants increased their insight into their patterns of self-criticism, emotional responses and identified areas where self-compassion can be developed. |
| | 1.4. Offered ways to manage difficult inner experiences | CFT provided tools and perspectives for managing self-criticism and intense emotions, helping participants externalise and reframe their inner experiences. |
| | 2. **Challenges engaging with particular CFT content** | Struggling with specific CFT exercises or concepts, such as compassionate letter-writing or compassionate self imagery. |
| How did the staff experience working on CFT concepts as part of a group? | 1. **Universality** | Being in a group contributed to the realisation others share similar challenges and that suffering is a shared human experience which fostered a sense of connection and reduced feelings of isolation. |
| | 2. **Learning with and from each other** | Participants widened their perspectives on challenging experiences by engaging with diverse viewpoints |
| If any, what was the impact of being part of the group on their well-being? | 1. **Cultivated compassion** | The group positively impacted participants' well-being by fostering self-compassion. |
| | 1.1. Recognising the need for self-compassion | Participants became aware of their need for self-compassion, realising its importance in their overall well-being. |
| | 1.2. Feeling deserving of self-compassion | Participants felt they had "permission" to prioritise their own needs without guilt, leading to behavioural changes that supported their well-being. |
| | 1.3. Responding with self-compassion | Participants developed a more compassionate inner dialogue, which helped reduce distress and manage difficult situations and self-criticism. |
| | 2. **Acquired skills to support well-being** | Practical skills, such as breathing exercises, were adopted to support ongoing emotional health and manage stress. |
| | 3. **Emotional improvement** | The group experience led to positive shifts in mood. |
| | 4. **Group as the start of an ongoing therapeutic journey** | The group was seen as an initial step towards further personal growth, often prompting continued therapy or self-improvement efforts. |
| What are the participants' views on improving the experience of CFT groups in the future? | 1. **Future delivery** | Elements that participants appreciated and suggestions for future groups. |
| | 1.1. Adapting content | Participants appreciated flexible content and hands-on learning but suggested more group interactions and inclusivity for varying needs. |
| | 1.2. Further support | Suggestions about periodic check-ins, follow-up emails, and one-on-one sessions to reinforce learning and maintain progress. |
| | 1.3. Supporting group safeness | Enhancing the sense of safeness through protected time, clear expectations, and detailed information about the group process. |
| | 1.4. Set up details | Smaller group sizes, a comfortable environment, same facilitator throughout were appreciated. |

judgment were prevalent. For example, Participant 2 mentioned, "*the beginning is quite daunting because you are feeling like that before you talk, you're like 'I'm going to share my life with these strangers here,' isn't it?*".

*Unexpected challenges.* Four participants encountered unexpected challenges, such as triggering unresolved emotions they thought they had dealt with, resulting in feelings of unexpected distress. For example, Participant 6 remarked that a group practice "*opened a can of worms*" and that it "*took me to a dark place that I didn't really want to go, but felt I needed to, if that makes sense*".

2. How did the staff experience learning about different concepts of CFT?

**Usefulness of engaging with key CFT content.** All participants found learning about CFT content useful, and engaging with it was not only informative but also transformative. Participants appreciated how CFT content helped them understand their personal experiences through accessible frameworks. They referred to several concepts, such as the three emotion regulation systems, the flows of compassion, and the evolutionary underpinnings of brain functions.

*Helped make sense of personal experiences.* Six participants found comfort in how CFT normalised their emotional responses. For instance, Participant 3 appreciated understanding brain functions, saying, "*Just understanding that, I particularly found that really helpful*", which eased their anxiety. Half of the participants noted that CFT content reduced feelings of self-blame. Participant 1 found the concept of "*we all start as this blank canvas*" valuable for understanding that personal development is shaped by early environments and helped them "*try not to punish*" themselves while Participant 6 noted, "*just having that insight, I guess to be able to go actually, this isn't my fault*" was particularly helpful. Six participants understood their difficulties through identifying blocks to self-compassion. For example, Participant 4 feared that being self-compassionate would mean "*letting yourself off the hook*", and Participant 8 viewed "*some of that self-compassion as selfish*".

Engaging with CFT led to shifts in participants' understanding of self-criticism. Participant 4 noted, "*It isn't a just response to what's happened*". Participant 7 found it "*really interesting*" to learn about the critical self's role as a protective mechanism in social dynamics, realising it helps avoid group rejection. Participant 8 viewed their critical self as a "*very human way of dealing with things*", while Participants 5 and 6 saw it as stemming from "*natural brain functions*" and their upbringing. Participant 4 recognised self-critical thoughts as cues for unmet needs, stating, "*Understanding what that part needs, what's it trying to tell you*" was beneficial. Participant 7 reframed their critical self as a "*defender*", understanding that it served a protective purpose rather than being a "*horrible villain*".

*Building a helpful definition of self-compassion.* Five participants found that learning about CFT concepts deepened their understanding of compassion. Four participants contrasted their previous views with a more nuanced understanding, leading to greater acceptance of self-compassion. Participant 2 realised that compassion involves more than just meditation; it includes self-reflection on feelings and reactions, allowing to "*give ourselves a break*". Participant 4 echoed this, emphasising the importance of recognising and meeting one's own needs. Participant 3 discovered that compassion is more than "*bubble baths*" and "*pink, fluffy*" ideas; "*compassion can be strong and standing up for yourself. And that was like, wow, never even thought of it like that*".

Four participants mentioned specific mental images of their compassionate self. For example, Participant 4 imagined a wiser version of themselves with boundaries, recognising their own needs whereas, for Participant 7, the compassionate self was like a "*light bird*" that helps avoid internalising others' judgments.

*Increasing self-awareness.* Almost all participants referred to having realisations about their ingrained patterns of self-criticism, the challenge of extending compassion to themselves and prioritising their own needs. Participant 2 highlighted the difficulty in "*allocating time to take care of ourselves*", especially in healthcare roles, and the struggle to permit themselves time off, echoed by Participant 7. Participant 5 had an epiphany about their lack of self-compassion, stating, "*Realising that I didn't have any compassion for myself whatsoever was quite difficult*", as they became aware of their "*really strong, really nasty*" inner critic. Participant 3 traced their critical voice to a parental figure, acknowledging how it

made them push "*past my own health to help other people*" These insights were seen as crucial steps towards self-awareness and change.

***Offered ways to manage difficult inner experiences.*** Seven participants found that CFT provided practical tools and new perspectives for managing difficult inner experiences, especially self-criticism and intense emotions. Six participants valued "*putting a name*" to their inner experiences, which helped them develop effective responses and adopt an observer stance. Participant 3 appreciated how CFT "*demystified*" emotions, providing language for previously abstract feelings, and making their critical self less intimidating. Participant 4 echoed this sentiment, as they started viewing their self-criticism as part of their "*threat system*", which offered "*some separation from you and how you feel, creating a helpful distance from it*" allowing them to see it as an understandable reaction rather than a personal flaw. Participant 7 also benefited from naming their inner critic, which helped them view it as "*a part of me, but it's not that big a part of me*" and described how they learned to "*interact*" with their self-critical side more healthily: "*I could effectively separate, that's that really defensive, self-critical side to my compassionate side and have them talk in like a really open way*". Participant 6 saw the exercises as a means to "*question and identify*" their internal states, reducing automatic self-criticism. Participant 8 recognised that their critical thoughts were often influenced by external voices, such as their parents, which helped them see these thoughts as "*just passing thoughts rather than actually who you are*".

**Challenges engaging with particular CFT content.** Six participants encountered challenges with specific CFT content, often struggling to connect personally with certain exercises or concepts. Despite these difficulties, they recognised the struggles as part of the therapeutic process. Participant 2 and Participant 8 had trouble forming comforting compassionate self-images. Participant 2 said, "*everyone that was popping into my head was being judged*", which made the process slow and difficult, while Participant 8 struggled to view their compassionate self positively, feeling it was "*a really selfish person*", which was upsetting.

Compassionate letter-writing exercises also posed difficulties for two participants. Participant 4 experienced a "*proper meltdown*" while writing to their younger self, finding it "*incredibly triggering*". Participant 5 was sceptical about the effectiveness of the exercise and found it "*quite tough*" as at the time they thought that "*I don't need to have compassion towards myself because I never have*", indicating resistance to offering themselves compassion stemming from long-held beliefs.

Addressing shame and self-criticism was another challenge. Participant 4 found it hard to "*get rid of shame*", noting that while they had tools to deal with other aspects, this was much harder to address. Participants 5 and 6 found exploring the roots of self-criticism tough, noting respectively that it "*was really tough because it was then starting to think about where it had come from*" and that it was "*by far the hardest week, by far the hardest activity*".

3. How did the staff experience working on CFT concepts as part of a group?

**Universality.** All participants described a profound sense of universality – the realisation that they were not alone in their struggles and that human suffering is a shared experience. This shared understanding arose from hearing others' experiences and openly sharing their own, fostering connection and reducing feelings of isolation. For instance, Participant 1 reflected, "*when things happen to you, you always think you're the only one. But actually, when you're in a group and you're talking openly, you realise you're not. You're not at all.*" Participant 5 shared a similar sentiment, describing the experience as, "*you sort of knew you were in it together*", while Participant 8 added that they valued "*to be seen and not to be alone*". Four participants highlighted how the group experience illuminated the commonality of human suffering, even when their specific challenges differed. Participant 2 noted, "*there's a common emotion in all of that, even if the situation is different. We share the same suffering.*" Participant 8 noted that seeing others struggle, despite appearing to cope well externally, made it clear that "*it's a real human natural response.*"

**Learning with and from each other.** All participants highlighted the value of engaging with CFT in a group. Seven participants emphasised the benefits of hearing diverse perspectives. For example, Participant 2 noted how hearing others'

experiences "*gives you a bigger range of experience*", allowed them to step back from their self-criticism and become "*a bit more of an observer rather than fused with that idea of yourself potentially*". Participant 4 added that listening to others' perspectives allowed them to "*forgive ourselves a little bit or myself, from what, how I was thinking or how I was behaving*".

Two participants also noticed that channelling their compassion towards others helped them cultivate self-compassion. Participant 4 reported "*You can connect to a sense of compassion for somebody else easier than you can connect to a sense of self-compassion*" adding that "*it's almost like you could kind of want to just get rid of that for them [...] And then it's like [...] that 'well, maybe I could do some of that myself'*". Similarly, Participant 6 found that being in a group supported them to extend the compassion they felt for group members to themselves, "*It was then easier, I guess, to kind of swivel it around to be able to talk to myself like I would to another member of the group*". This group dynamic helped participants reframe their internal dialogue.

Learning in a group also facilitated a better understanding of the content. Participant 1 experienced "*light bulb moments*" from others' contributions and Participant 5 found that when they did not understand part of the content, someone else's explanation often made it clearer. This collaborative learning environment was instrumental in making the CFT concepts more accessible and relatable.

4. If any, what was the impact of being part of the group on their well-being?

**Cultivated compassion.** Being part of the group had a positive impact on participants' well-being by cultivating a sense of self-compassion.

*Recognising the need for self-compassion.* Four participants realised through the group how much they needed to cultivate self-compassion and saw that as a key change that came out of the group. Participant 5 noted that "*Recognising that I don't do it was a big thing*", while Participant 1 reflected, "*I tend to put other people first, where actually I need to put myself first.*" Participant 3 reinforced this by recognising that they must care for themselves to support others, comparing it to the "*aeroplane oxygen mask*" analogy, saying, "*In order to really look after the ones around me, I need to look after myself.*" This was also echoed by Participant 2 who also said, "*I'll be better to everyone else if I'm OK.*"

*Feeling Deserving of Self-Compassion.* All participants developed their sense of deserving self-compassion, which often led to behavioural changes prioritising their well-being and addressing their needs.

Participant 2 said the group "*gave me some strength to claim it*" without feeling guilty, "*I can take some time to myself without feeling the worst person in the world*". Participant 3 shared that their "*biggest sort of take away*" was that "*standing up for myself is OK*" and started to say no without feeling "*selfish".* This newfound sense of deserving self-compassion also transpired to "*develop sort of almost like little rituals at home like a little evening ritual before I go to bed and. And I find that helpful.*" These made them "*feel good and calm*" and reduced some of their day-to-day stress. Participant 4 expressed similar sentiments, noting that it is "*all right to have needs and then it's alright to kind of work towards meeting them as well*" which led to investing in self-care daily routines: "*It's just an example of like having enough worth to think that one should do that.*" For Participant 5, a key change was allowing themselves to seek help*: "It was almost OK to get help and start to think about these things*".

*Responding with Self-Compassion.* Seven participants referred to new ways of responding to difficult situations and inner experiences with compassion. Participants 1, 2, 5, 6 and 7 spoke about having developed a more compassionate self-talk. For example, Participant 5 said that "*there's that voice in my head, sometimes it just says take it easy today, you've got a day off, just relax, you deserve that.*" Participant 6 highlighted that the group "*has helped me massively with myself [...] the way that I talk to myself has been amazing*".

For four participants, their more compassionate inner voice would also question and challenge their critical thinking. For example, Participant 6 spoke about how they learned to question their critical thinking by saying "*would I say that to my best friend? [...] So then why am I saying it to myself?*" and how they noticed "*questioning myself a lot now, which I never*

*did before. You just believed what your head says*" which they noted reduced self-criticism. For Participant 7, their newly developed inner self-compassionate talk would support them detach from others' negative feedback, by saying "*It's OK to rant and let it go*". They would step back and think, "*If I was in the group, what would other people say to me? Probably not what I'm saying to myself right now.*"

**Acquired skills to support well-being.** All participants also spoke about how they acquired tools, via skills-based learning, that they use to support their well-being after the group ended.

Four participants found breathing exercises useful. Participant 5 shared that it helps when feeling overwhelmed: "*Breathing completely slows me down and takes me out of it*" echoed by Participant 4 who found them "*very grounding.*" Participant 7 highlighted its role in connecting with compassion, saying, "*It's a very physical way of letting go of things*" while Participant 6 noted different patterns for specific needs: "*To wake up, you do big in-breath; to calm down, long out-breaths […] learning those differences actually is a big help*".

Participant 8 learned to adjust their body posture to reduce anxiety: "*I recognise anxiety in my body and try to change my body to reduce that anxiety, like lowering my shoulders.*" Two participants kept using the safe space visualisation to feel calmer, with Participant 5 saying*, "It completely calms my body"*. Two participants mentioned using meditation. For instance, Participant 1 said they use it when needed: "*If I feel a situation, I will go and do it.*"

**Emotional improvement.** Several participants reported notable improvement in their emotional well-being. For example, Participant 2 described the group as a crucial factor in breaking a cycle of depression and low mood: "*I was in a very dark place...going to the group every week, it broke the cycle.*" This change in emotional state was significant enough that they no longer felt the need to take time off from work, indicating a considerable improvement in their mental health. Participant 6 echoed this sentiment, noting that while they went through a "*dark place*" during the process, the group ultimately provided them with strategies to navigate through it. Participant 7 noticed a positive shift in their behaviour and interactions, sharing, "*I've noticed that I'm smiling a lot more again, which is really nice*". They also started making jokes and social plans, activities they had previously avoided. Participant 4 highlighted the development of a support network as a positive effect on their well-being, stating that "*having new friends is literally a positive effect of my well-being.*"

**Group as the start of an ongoing therapeutic journey.** Five participants perceived the group as the starting point of their therapeutic journey and catalyst for further personal growth. Participant 2 noted it highlighted the need for more one-on-one work, while Participant 4 described it as "*fantastic groundwork*" for their current trauma-focused therapy. Participant 5 reflected that the group was "*the start*" of their therapeutic journey, realising it was "*OK to get help*" and Participant 8 viewed it as a "*gateway*" that allowed them to do further therapy. Participant 6 pointed out that: "*it's ongoing work even when you finish the group*", recognising that therapy is not about a final destination.

5. What are the participants' views on improving the experience of CFT groups in the future?

**Future delivery: *Adapting content.*** Participants valued the flexibility in content delivery, hands-on learning, and the balance between structured and participant-led activities. For example, Participant 8 remarked, "*It was really lovely because it met our needs then perfectly*", describing the experience as "*bespoke*". Similarly, Participant 7 positively commented on how the intervention would "*put the group's needs ahead of the group's strict format*"*.*

While the group aimed to be flexible, Participant 3 suggested the need for more group discussion. Participant 8 highlighted the need to adapt exercises for varying physical abilities, noting that some might "*feel very differently about their body*". Participant 4 noted that some foundational work might be necessary before engaging in certain exercises: "*Perhaps some groundwork on some of the exercises as they might be challenging, and that's normal, and this is what we do with it if they are*".

Experiential learning, through practical exercises like letter writing and meditation, was highly valued. Four participants found these activities helped externalise their thoughts and engage actively with the material. For example, Participant 7

noted that "*experiencing a technique first-hand rather than just watch someone else do it*" and Participant 3 appreciated being involved: "*You're actually, you know, participating in something.*"

**Further support.** While participants valued the ability to reach out to the facilitator, they desired more structured and consistent follow-up to maintain their progress and stay connected to the practices. Participant 1 suggested that periodic check-ins would be beneficial. They proposed "*a 6-monthly catch-up*" to refresh course lessons amid their busy lives. Similarly, Participant 4 recommended follow-up emails as reminders of key exercises, while Participant 3 proposed "*booster sessions*" to revisit course material with the facilitator, "*like a COVID jab*". Other suggestions included individual sessions during or after the group to provide personalised support if needed. For example, Participant 2 proposed one-on-one sessions, either during or after the course, to explore further moments that may trigger deeper emotions: "*Maybe there will be sessions, moments that will trigger something...it would be positive to explore that*".

**Supporting group safeness.** Participants suggested ways to enhance safeness in the group. A key element was the concept of protected time, which five participants appreciated as it gave them permission to focus on their needs. Participant 2 compared this time to an unavoidable doctor's appointment, saying, "It *was labelled as treatment, as therapy, you need to go. You can't miss the doctor*". This framing helped them prioritize their well-being. Participant 7 echoed this sentiment: "*Just having that protected time to not think about work was really helpful*". Participants 5, 6, and 7 also noted that this time was valuable for practising skills they otherwise would not find time for.

To further enhance group safeness, some participants suggested setting clearer expectations about the therapy process. Participant 6 emphasised the need to communicate that therapy requires continued personal effort and is not about becoming "*100% happy, 100% of the time.*" Participant 2 found it helpful to know that "*the goal of the group is about compassion, not solving everything in your life.*" Participant 4 and 6 recommended more upfront information about the group process to ease concerns about participation and provide reassurance that participants do not have to share more than they are comfortable with.

**Group set-up.** Participants highlighted practical elements of the group they valued and suggested improvements. A smaller group size was widely appreciated for fostering intimacy and comfort, as mentioned by five participants. Specifically, Participant 6 felt "*five or six would have been big enough*" while Participant 8 viewed a group of four as "*a really lovely size*".

Comfort in the environment was important, with three participants suggesting the need for a quieter setting. Participant 8 mentioned the benefit of using the same room to create a familiar space. Participant 6 also valued having the same facilitator throughout the group as it helped feel supported in emotionally challenging moments.

## Discussion

This study explored the experiences of NHS staff in a CFT group, revealing positive outcomes across several areas. Participants valued the group's ability to foster a sense of safeness. Engaging with CFT content increased self-awareness and helped participants understand their inner experiences through evolutionary psychology frameworks, often reducing self-blame. This enabled them to develop healthier ways of relating to inner experiences. A key outcome was the cultivation of self-compassion, which motivated self-compassionate actions and led to improved well-being. While participants valued the flexible group content and experiential learning, some found certain exercises challenging, indicating that tailored support may be needed alongside the group format.

Participants reported noticeable improvements in self-compassion, attributing them to the group, corresponding to the core aim of CFT [10]. Broadly, participants described a journey from learning and then applying CFT concepts to develop a compassionate understanding of their difficulties, contributing to a reduction in feelings of self-blame, consistent with Gilbert's [11] theoretical account of CFT's role in reducing self-criticism by cultivating self-compassion. According to participants' accounts, a key element for the development of self-compassion was gaining a more accurate understanding of compassion, discarding previous misconceptions about it [18] and understanding FBRs as defences developed through

personal experiences and history [37,56]. Engaging in experiential exercises that involved separating and interacting with different parts of themselves helped participants externalise and create some distance from their critical self and build a sense of mind awareness that facilitated changes in self-to-self relating, aligning with suggestions on effectively addressing FBRs [56,57].

Recognising the need for and cultivating compassion facilitated participants' ability to relate to difficult threat-based emotions and self-criticism in more constructive ways. Many participants referred to developing a compassionate self-image or inner voice, enabling them to respond to inner threat-based experiences with self-compassion, reflecting a core CFT process [58]. These findings are consistent with prior research indicating that CFT helps healthcare professionals transform their self-talk into one marked by self-compassion [59].

A developed sense of deserving compassion was evidenced by participants' intention and day-to-day actions aiming to meet their emotional needs, such as setting healthy boundaries in relationships, allowing themselves to rest, and using CFT practices that supported their well-being. This seems to complement recent quantitative research indicating that CFT is effective in reducing overall negative mental health outcomes and enhancing compassion for both self and others [60]. Other quantitative research has also linked increases in self-compassion to improved mental health outcomes [61], including reductions in stress, depression, and anxiety [16].

The group format was viewed as fundamental to the therapeutic process, consistent with previous research [58,62]. Group therapy's unique ability to address self-criticism and shame through affiliative and caring relationships [45,63] was echoed by participants and is supported by other CFT studies [64]. The development of 'group safeness' was reported as a central construct, fostering engagement and therapeutic change. It allowed participants to give and receive compassion from others, which in turn strengthened self-compassion [65,66], as participants reported that directing compassion towards others and learning from others' experiences and feedback helped them develop self-compassion. The affiliative nature of the group, which was strengthened by their shared professional NHS identity, seemed to enhance participants' ability to access, tolerate, and direct affiliative motives and emotions towards themselves, corroborated by a recent study on group-based CFT [67].

## Clinical implications

Findings from this study offer meaningful clinical implications. The CFT group was perceived as acceptable for NHS staff who reported pervasive self-criticism, regardless of their underlying mental health difficulties, in line with previous research [29]. For some, the group served as an early therapeutic step or as a stabilisation intervention prior to trauma-focused therapy, aligning with recommendations suggesting CFT's utility in early PTSD treatment where factors of shame or internalised criticism may impede processing [68,69]. Thus, CFT seemed useful for NHS staff with this clinical profile.

One of the most valued aspects enhancing group safeness and engagement was the provision of protected time to attend sessions. This agrees with previous research emphasising the role of organisational support for healthcare professionals engaging in CFT [24,32]. Given the tendency of healthcare professionals to prioritise others' needs over their own [24,70,71] and the potential need for permission to attend to their own needs [25], offering dedicated time during work hours can provide both practical and psychological permission to focus on their well-being.

Co-occurring one-on-one sessions were suggested for participants who might need further guidance on managing emotional triggers and addressing FBRs to compassion. Participants might struggle to hold on to a lasting compassionate image [72] or generate an image that also has self-critical features [73]. McManus et al. [74] suggest that due to the threat system activation, such difficulties are likely inevitable in CFT, and harder to manage in a group. Therefore, opportunities for individualised support may be necessary.

To sustain progress post-group, follow-up sessions, email reminders, and booster sessions were recommended, echoing findings of the enhanced retention of therapeutic content and skills in therapeutic interventions [67,75,76]. Booster

sessions could strengthen social bonds and group cohesion, motivating continued CFT practice [45], and thus may be considered in the design of similar groups.

## Limitations and future directions

This study was subject to several limitations. The findings should be interpreted cautiously, as the study involved a small, homogenous sample of self-selected participants (eight out of 20 eligible participants) consisting only of White females. This may limit applicability of the findings to the wider NHS workforce, given that women, while comprising approximately 74% of NHS staff [77], were exclusively represented in this sample, and around 25% of NHS staff identify as belonging to ethnic minority groups [78] who were not represented. Such demographic and contextual factors may influence how the intervention is experienced and constrain transferability both within this Trust and to other NHS settings. However, certain aspects of the findings may be relevant to other NHS contexts, such as the shared healthcare professional identity and workplace pressures that appeared to enhance group cohesion and might be common across many NHS environments. Additionally, the emphasis on 'protected time' and organisational support in this study is consistent with findings from other NHS wellbeing interventions [79]. Future research could explore CFT group delivery across more diverse staff populations, a wider range of NHS roles, and different care settings to better understand the extent and boundaries of transferability.

Furthermore, self-selection bias and the inclusion criterion of attending at least 8 of 12 sessions may have resulted in an overrepresentation of favourable experiences and an underrepresentation of those who attended fewer sessions, disengaged or chose not to participate [80]. Nonetheless, critical and disconfirming views expressed by participants were incorporated into the analysis to ensure that both positive and challenging aspects of the intervention were reflected in the findings.

The high attendance rate with 15 out of 20 participants attending eight or more sessions suggests the intervention was accessible and manageable for most, and it also provides additional evidence to its acceptability. Investigating the reasons for dropout could help fine-tune the intervention to increase both its feasibility and acceptability and to better understand how to support staff more effectively. Prior research on the acceptability and feasibility of a group-based CFT with inpatient mental health staff highlights organisational barriers that impede participation [24], suggesting that exploring similar challenges here could inform implementation improvements.

Although this study was focused on the intervention experiences of NHS staff at an acute hospital Trust, it provides insights for future research on compassion-based interventions for healthcare professionals. So far, research has predominantly focused on self-compassion programs, primarily aiming to reduce burnout, enhance resilience, and improve general workplace well-being [81], varying significantly in duration and content [24,25,32,82,83], limiting broader applicability. While these interventions have shown potential in enhancing self-compassion and reducing burnout [84], the focus on group-based CFT for staff with mental health problems underpinned by high self-criticism addresses a distinct need. Future research should use larger samples of healthcare professionals, adopting quantitative or mixed methods to assess the intervention's effectiveness more comprehensively. Using quantitative measures could also help determine whether any observed improvements in mental health outcomes are directly attributable to changes in compassion-related factors or whether other factors are contributing to these changes.

Finally, while the analysis primarily followed an inductive approach, it was structured around predefined evaluation questions and incorporated some deductive coding based on CFT theory. This may have constrained the development of broader, integrative themes and potentially directed attention toward aspects most relevant to the evaluation aims. Adopting a fully inductive thematic analysis approach in future projects could yield richer insights into participants' experiences, such as contextual barriers and facilitators not captured by the evaluation framework, or broader impacts of the intervention beyond the predefined questions of interest.

## Conclusions

This study highlights the value of group-based CFT for NHS staff, showing its potential to foster self-compassion and improve well-being, within the context of mental health difficulties. The group format was identified as a key factor in creating a sense of safeness and belonging, enhancing the therapeutic experience by allowing participants to give and receive compassion within a supportive environment. However, further individual support may be necessary for those who need additional assistance with emotional challenges. Future studies should expand the research with larger, more diverse samples and investigate long-term outcomes to fully understand CFT's impact on healthcare professionals.

## Supporting information

**S1 File. Interview schedule.**
(DOCX)

**S1 Table. Consolidated criteria for reporting qualitative research checklist.**
(DOCX)

**S2 Table. Thematic analysis table presenting number of participants mentioning theme and subtheme by research question.**
(DOCX)

## Acknowledgments

We are grateful to the NHS staff who participated for their time. The authors have declared that there are no conflicts of interest in relation to the subject of this study. This work was undertaken as part of AR's Clinical Psychology Doctorate at the University of East Anglia.

## Author contributions

**Conceptualization:** Aikaterini Rammou, James Baker.

**Data curation:** Aikaterini Rammou, James Baker.

**Formal analysis:** Aikaterini Rammou, Sophie M. Allan, Martha Dean-Tozer, James Baker.

**Investigation:** Aikaterini Rammou, James Baker.

**Methodology:** Aikaterini Rammou, James Baker.

**Project administration:** Aikaterini Rammou.

**Resources:** Aikaterini Rammou, James Baker.

**Supervision:** Sophie M. Allan, James Baker.

**Writing – original draft:** Aikaterini Rammou.

**Writing – review & editing:** Aikaterini Rammou, Sophie M. Allan, Martha Dean-Tozer, James Baker.

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
