## [Decision Letter · Decision Letter 0]

8 Jul 2025

PONE-D-25-22995A Qualitative Evaluation of a Compassion-Focused Therapy Group Intervention for UK Healthcare Staff at an Acute Hospital TrustPLOS ONE

Dear Dr. Rammou,

Thank you for submitting your manuscript to PLOS ONE. After careful consideration, we feel that it has merit but does not fully meet PLOS ONE’s publication criteria as it currently stands. Therefore, we invite you to submit a revised version of the manuscript that addresses the points raised during the review process.

**ACADEMIC EDITOR:** 

Concerns:

Discuss more clearly how the sample (all White, female, self-selected staff) limits the generalisability of the findings.Clarify why pre-intervention quantitative measures (PHQ-9, GAD-7, FSCRS) are included and what they add to the study. If they are only descriptive, explain that clearly.Strengthen the reflexivity section by more critically examining how the prior supervisory relationship between interviewer and group facilitator might have influenced data collection and analysis.Include more attention to any negative, critical, or disconfirming participant views, and discuss these in the results and discussion.Expand discussion of transferability—explain how these findings might (or might not) apply to other NHS settings, staff groups, or populations.Clarify the process for external researchers to request access to the data held by the NHS Trust, or specify any restrictions.Explain more transparently how inductive coding was balanced with using pre-defined evaluation questions, and discuss any limitations this imposed.Consider adding a simple flow diagram or table to make participant recruitment and inclusion clearer (20 invited, 15 completed enough sessions, 8 interviewed).Improve clarity in the reporting of participant quotes by avoiding repetition and grouping similar themes where possible.Discuss confidentiality concerns given the small group size and in-house nature of the service, and how these were managed.

We look forward to receiving your revised manuscript.

Kind regards,

Mohd Ismail Ibrahim, MCom.Med

Academic Editor

PLOS ONE

Journal Requirements:

Reviewers' comments:

Reviewer's Responses to Questions

**Comments to the Author**

1. Is the manuscript technically sound, and do the data support the conclusions?

Reviewer #1: Yes

2. Has the statistical analysis been performed appropriately and rigorously? 

Reviewer #1: I Don't Know

3. Have the authors made all data underlying the findings in their manuscript fully available?

Reviewer #1: Yes

4. Is the manuscript presented in an intelligible fashion and written in standard English?

Reviewer #1: Yes

5. Review Comments to the Author

Reviewer #1: Thank you for asking me to review the article and I thank the authors for a comprehensive and easy to read manuscript that covers all aspects expected in a qualitative research article.

Technical comments:

• It was unclear how did the authors formulate and came to their interview questions. Specifically, it was unclear what theory informed the interview questions as these were specific and detailed. I could see that coding was not subjected to a particular conceptual framework (as stated in the data analysis section) so what theory guided the data analysis and interview questions exactly? Afterall, this is a research paper and the expectation should be that some sort of a theory guided the research.

If the paper was a result of an NHS commissioned study to evaluate the therapy used then the authors should make that clear in the paper and to the editor who should make the decision on the paper’s rigor.

There are many research articles that discuss CFT therapy and its use in different settings so one should expect to see some sort of a theory behind its use, application and effectiveness in clinical settings or nonclinical settings.

Non-technical comments:

I thank the authors for a comprehensive easy-to-read paper but noted some few easy-to fix edits where appropriate to improve the paper:

Abstract:

• No background information upfront that introduces the reader on why this study was conducted or why is it needed (line 21). This can be added in a sentence upfront.

Introduction:

• (NHS England, 2016), is this meant to be a reference? (line 47). If yes, convert to PLOS One style.

• Please add page number after quotes in the in-text format as follows [9(p.?)]. (line 53)

Methods:

• ‘Were invited’ (line 135). Were also invited?

• Guidelines recommend (not recommending) (line 154)

Measures:

• 14th of January (line 199)

Discussion:

• In line with what? Please specify (line 580)

• The reference to chair work in group sessions as an activity (lines 583-584) was not mentioned or reflected in any of participants quotes. Suggest either reflect in participants quotes or replace with another technique mentioned by participants in the results section so they align.

6. PLOS authors have the option to publish the peer review history of their article (what does this mean? ). If published, this will include your full peer review and any attached files.

**Do you want your identity to be public for this peer review?** For information about this choice, including consent withdrawal, please see our Privacy Policy .

Reviewer #1: **Yes: ** Tamara Al-Obaidi (PhD)

---

## [Author Response · Author response to Decision Letter 1]

5 Sep 2025

We thank the Academic Editor and Reviewer for their thoughtful and constructive feedback, which has helped us to strengthen the clarity, transparency, and rigour of our manuscript.

Please find our responses below to each issue raised. A copy of this has also been submitted in the "Response to Reviewers" file as requested.

Response to the Academic Editor

• Discuss more clearly how the sample (all White, female, self-selected staff) limits the generalisability of the findings.

We have expanded the discussion of sample limitations to address this point.

• Clarify why pre-intervention quantitative measures (PHQ-9, GAD-7, FSCRS) are included and what they add to the study. If they are only descriptive, explain that clearly.

We have clarified in the Methods that the pre-intervention PHQ-9, GAD-7, and FSCRS scores were included for descriptive purposes only. They provide contextual information about participants’ baseline levels of depression, anxiety, and self-criticism, which helps to characterise the sample but were not used for statistical comparison or hypothesis testing.

• Strengthen the reflexivity section by more critically examining how the prior supervisory relationship between interviewer and group facilitator might have influenced data collection and analysis.

We have expanded the reflexivity section to explicitly address how the prior supervisory relationship between the interviewer and group facilitator could have influenced data collection and interpretation, and described the specific steps taken to mitigate potential bias.

• Include more attention to any negative, critical, or disconfirming participant views, and discuss these in the results and discussion.

We have carefully considered and incorporated all critical or disconfirming participant views in the analysis. These have been reflected particularly in the sections summarising participants’ suggestions for improvement and in subthemes referring to challenges faced when taking part in the group. However, we acknowledge that the inclusion criterion requiring attendance at 8 or more of the 12 sessions may have meant that our sample was composed of individuals who found the intervention beneficial or were more positively predisposed toward it, which may partly explain the relative absence of strongly negative feedback. This limitation has now been explicitly noted in the Discussion.

• Expand discussion of transferability—explain how these findings might (or might not) apply to other NHS settings, staff groups, or populations.

We have expanded the discussion of transferability in the Limitations by clarifying how demographic/contextual factors may limit applicability beyond (and within) our acute hospital trust setting, and by noting where findings may be relevant across NHS contexts (e.g., shared caregiving identity, workplace pressures

• Clarify the process for external researchers to request access to the data held by the NHS Trust, or specify any restrictions.

Data cannot be shared publicly because it contains information collected under NHS service evaluation governance and is subject to the UK Data Protection Act (2018) and the Trust’s confidentiality policy. External researchers may submit a formal request to the NHS Trust’s Clinical Audit department (ClinicalAudit@wsh.nhs.uk) which would be reviewed by the Trust’s Information Governance team and, if appropriate, approved by the Caldicott Guardian before any anonymised data could be shared. The Data Availability statement in the submission form has been updated accordingly.

• Explain more transparently how inductive coding was balanced with using pre-defined evaluation questions, and discuss any limitations this imposed.

We have clarified in the Methods section how inductive coding was balanced with the predefined evaluation questions, specifying that coding began inductively from participants’ accounts but that the grouping of codes was also informed by the evaluation questions and deductive insights from CFT theory. In the Discussion, we have added a statement on the limitations this imposed, noting that this approach may have constrained the development of broader, integrative themes and directed attention toward aspects most relevant to the evaluation aims.

• Consider adding a simple flow diagram or table to make participant recruitment and inclusion clearer (20 invited, 15 completed enough sessions, 8 interviewed).

Thank you for this suggestion. A recruitment flow diagram has been added for clarity in Method under the Participants subsection.

• Improve clarity in the reporting of participant quotes by avoiding repetition and grouping similar themes where possible.

We appreciate the reviewer’s suggestion and have carefully revised the Results section to enhance clarity. While we have retained the original thematic structure, as it reflects the analytic process and was grounded in participants’ accounts, we have reduced repetition by streamlining the participant quotes within each theme and subtheme. Where several quotes conveyed the same core idea, we kept the most representative examples and retained additional quotes only when they offered a meaningful nuance or different perspective. This approach preserves the richness and authenticity of participants’ voices while improving readability.

• Discuss confidentiality concerns given the small group size and in-house nature of the service, and how these were managed.

We have added a section on “Confidentiality” after the Ethics statement describing how confidentiality was maintained in the context of small, in-house groups. This includes anonymising transcripts and quotes, removing potentially identifying information, and aggregating demographic data where necessary to protect participant identity.

Response to the Reviewer

Technical comments:

• It was unclear how did the authors formulate and came to their interview questions. Specifically, it was unclear what theory informed the interview questions as these were specific and detailed.

I could see that coding was not subjected to a particular conceptual framework (as stated in the data analysis section) so what theory guided the data analysis and interview questions exactly? Afterall, this is a research paper and the expectation should be that some sort of a theory guided the research.

If the paper was a result of an NHS commissioned study to evaluate the therapy used then the authors should make that clear in the paper and to the editor who should make the decision on the paper’s rigor.

There are many research articles that discuss CFT therapy and its use in different settings so one should expect to see some sort of a theory behind its use, application and effectiveness in clinical settings or nonclinical settings.

We sincerely thank the reviewer for this valuable methodological question regarding the theoretical foundations of our interview questions and analysis approach. We appreciate the opportunity to clarify our methodological positioning and to provide further detail on our research design decisions.

This study was conducted within a pragmatic qualitative framework, with the aim of capturing participants’ lived experiences of the CFT group in order to inform service development and assess acceptability. A qualitative methodology was chosen to explore these subjective experiences, reflecting Braun and Clarke’s (2014) description of its value for accessing “rich and compelling insights into the real worlds, experiences, and perspectives of patients and health care professionals” (p. 1).

The semi-structured interview schedule was collaboratively developed by the group facilitator (a clinical psychologist) and the main researcher. Its design was guided by:

1. CFT theory – to explore participants’ experiences of core CFT concepts central to the intervention delivered and their relevance to changes in understanding and managing inner experiences.

2. Group therapy process literature – drawing on established constructs such interpersonal learning, inquiring about this in this particular context which has not been researched.

3. Intervention evaluation principles – with a focus on acceptability, perceived impact, and recommendations for improvement.

These clarifications have now been incorporated into the Methods section of the manuscript with their corresponding references from the literature.

With regard to analysis, although coding was primarily inductive, we used a combination of inductive and deductive reflexive thematic analysis (Braun & Clarke, 2022). Coding began without a predetermined framework, allowing themes to be constructed directly from participants’ accounts (inductive), while interpretation was subsequently informed by CFT theory when situating the findings within the existing evidence base (deductive). This has been explicitly clarified in the Data Analysis section.

Finally, in response to the reviewer’s query on commissioning, we confirm that the CFT groups were delivered as part of routine clinical practice. The evaluation was undertaken as part of the main researcher’s doctoral training in clinical psychology and was not commissioned by the NHS Foundation Trust where the project took place.

We hope that these amendments and clarifications fully address the reviewer’s concerns, and we are grateful for the opportunity to strengthen the transparency and rigour of our manuscript.

Non-technical comments: I thank the authors for a comprehensive easy-to-read paper but noted some few easy-to fix edits where appropriate to improve the paper:

Abstract:

• No background information upfront that introduces the reader on why this study was conducted or why is it needed (line 21). This can be added in a sentence upfront.

We thank the reviewer for this suggestion. We have added background information at the beginning of the abstract to provide context for why this study was conducted and to highlight its relevance.

Introduction:

• (NHS England, 2016), is this meant to be a reference? (line 47). If yes, convert to PLOS One style.

Thank you for catching this formatting error. We have corrected the NHS England citation to proper PLOS ONE reference style.

• Please add page number after quotes in the in-text format as follows [9(p.?)]. (line 53)

We have added page numbers to all direct quotes following PLOS ONE citation style. Specifically, the one relating to 'antidote' was not a direct quote so the quotation marks were removed.

Methods:

• ‘Were invited’ (line 135). Were also invited?

Thank you for noting this clarity issue. We recognised that the original wording could make it difficult for readers to follow and potentially had obscured the fact that only staff meeting all listed criteria were invited to participate. We have revised the sentence to improve both readability and precision.

• Guidelines recommend (not recommending) (line 154)

Thank you for the grammatical correction. We have changed 'recommending' to 'recommend'.

Measures:

• 14th of January (line 199)

Thank you for spotting this. This error has now been rectified.

Discussion:

• In line with what? Please specify (line 580)

Thank you for highlighting this omission. Our intention was to link the finding of reduced self-blame to Gilbert’s (2009) theoretical account of CFT’s role in reducing self-criticism by cultivating self-compassion. We have revised the sentence to make this link explicit.

• The reference to chair work in group sessions as an activity (lines 583-584) was not mentioned or reflected in any of participants quotes. Suggest either reflect in participants quotes or replace with another technique mentioned by participants in the results section so they align.

We thank the reviewer for identifying this discrepancy. We have added a representative participant quote to the Results section under 'Offered Ways to Manage Difficult Inner Experiences' to better support our discussion point. We have also revised the Discussion to describe the technique based directly on participants’ accounts rather than using clinical terminology (“chair work”)

We hope the revisions made in response to each comment have addressed all concerns and improved the manuscript, and we appreciate the opportunity to resubmit our work for your consideration.

---

## [Decision Letter · Decision Letter 1]

17 Sep 2025

A qualitative evaluation of a compassion-focused therapy group intervention for UK healthcare staff at an acute hospital trust

PONE-D-25-22995R1

Dear Dr. Rammou,

We’re pleased to inform you that your manuscript has been judged scientifically suitable for publication and will be formally accepted for publication once it meets all outstanding technical requirements.

Kind regards,

Mohd Ismail Ibrahim, MCom.Med

Academic Editor

PLOS ONE

Additional Editor Comments (optional):

The authors have addressed all concerns raised previously. The revisions have enhanced the clarity, methodological transparency, and overall quality of the manuscript. I am satisfied with the changes made.

Reviewer #1:

Reviewers' comments:

Reviewer's Responses to Questions

**Comments to the Author**

1. If the authors have adequately addressed your comments raised in a previous round of review and you feel that this manuscript is now acceptable for publication, you may indicate that here to bypass the “Comments to the Author” section, enter your conflict of interest statement in the “Confidential to Editor” section, and submit your "Accept" recommendation.

Reviewer #1: All comments have been addressed

2. Is the manuscript technically sound, and do the data support the conclusions?

Reviewer #1: Yes

3. Has the statistical analysis been performed appropriately and rigorously? 

Reviewer #1: N/A

4. Have the authors made all data underlying the findings in their manuscript fully available?

Reviewer #1: Yes

5. Is the manuscript presented in an intelligible fashion and written in standard English?

Reviewer #1: Yes

6. Review Comments to the Author

Reviewer #1: I thank the authors for the effort in addressing our comments comprehensively and clearly in their statement as well as in the manuscript (where appropriate). I have no further concerns or comments.

7. PLOS authors have the option to publish the peer review history of their article (what does this mean? ). If published, this will include your full peer review and any attached files.

**Do you want your identity to be public for this peer review?** For information about this choice, including consent withdrawal, please see our Privacy Policy .

Reviewer #1: **Yes: ** Dr Tamara Al-Obaidi

---

## [Editor Report · Acceptance letter]

PONE-D-25-22995R1

PLOS ONE

Dear Dr. Rammou,

I'm pleased to inform you that your manuscript has been deemed suitable for publication in PLOS ONE. Congratulations! Your manuscript is now being handed over to our production team.

Kind regards,

on behalf of

Dr. Mohd Ismail Ibrahim

Academic Editor

PLOS ONE